# Prediction of Surface Roughness in Gas-Solid Two-Phase Abrasive Flow Machining Based on Multivariate Linear Equation

**DOI:** 10.3390/mi13101649

**Published:** 2022-09-30

**Authors:** Wenhua Wang, Wei Yuan, Jie Yu, Qianjian Guo, Shutong Chen, Xianhai Yang, Jianchen Cong

**Affiliations:** 1School of Mechanical Engineering, Shandong University of Technology, Zibo 255000, China; 2Tianrun Industrial Technology Co., Ltd., Weihai 264200, China

**Keywords:** gas-solid two-phase flow principle, abrasive pool machining, orthogonal experiment, surface roughness prediction model

## Abstract

The main purpose of this study is to explore a surface roughness prediction model of Gas-Solid Two-Phase Abrasive Flow Machining. In order to achieve the above purpose, an orthogonal experiment was carried out. Q235 steel as processing material and white corundum with different particle sizes as abrasive particles were used in the experiment. Shape and spindle speed were the main reference factors. The range method and factor trend graph are used to comprehensively analyze the experimental results of different processing stages of the detection point, and the optimal parameter combination of A_3_B_2_C_1_D_2_ was obtained. According to the experimental results, a multiple linear regression equation was established to predict the surface roughness, and the experimental results were solved and significantly analyzed by software to obtain a highly reliable prediction model. Through experiments, modeling and verification, it is known that the maximum error between the obtained model and the actual value is 0.339 μm and the average error is 0.00844 μm, which can better predict the surface roughness of the gas-solid two-phase flow abrasive pool.

## 1. Introduction

Abrasive Flow Machining (AFM) is a new finishing method for deburring and rounding workpieces by extruding fluid abrasives. The polishing technology was originally developed according to the deburring needs of aerospace parts in the US aerospace industry, mainly for the finishing of narrow slits, tiny holes and special-shaped holes. After successful research and development, its application range has gradually expanded [1]. Since the abrasive medium is fluid, AFM has a good application in micropores, complex pores and other pore structures [2,3,4,5]. In the process of AFM, the abrasive particles are relatively small, meaning less damage to the surface of the workpiece. The residual stress on the surface of the workpiece can be reduced, and the concentrated stress on the surface of the workpiece can be reduced while the machining quality is improved, and the wear resistance and corrosion resistance of the workpiece can be increased [6,7]. However, the fluid modulus used in some AFM technologies is usually viscous abrasives, which will encounter greater resistance when the abrasives move relative to each other, and the plasticity and cohesion of some polymer media will cause the abrasives to avoid the processing area, resulting in the surface processing quality of the material being poor. Therefore, in the field of abrasive pools, new grinding media can be developed, and the reduction in media cost, media sustainability, and waste generation still needs to be continuously explored [8].

In response to the exploration of new processing methods of AFM, Barletta [9] innovatively proposed the fluidized bed to assist the spindle-type tumbling finishing and found that the fluidized abrasive grains can reduce the energy required to move the workpiece, and the fluidized abrasive grains can machine the ductile metal with a better mirror finish; for the study of new abrasive media for AFM, Pham et al. [10] explored an abrasive air jet technology to process alloy materials, which reduced the surface roughness *Ra* by 40 times. The gas-solid two-phase flow abrasive flow processing proposed in this paper is to use high-speed airflow to accelerate the solid-phase abrasive particles to realize the relative motion between the gas-solid two-phase flow and the workpiece, so as to achieve the purpose of finishing (and solid phase abrasive particles are innovative).

However, the flow state of gas-solid two-phase flow, workpiece shape and abrasive grain shape will affect the machining quality of the workpiece surface, and the surface roughness depends on many machining parameters. Therefore, many researchers try to analyze and predict the roughness of the machined surface through modeling. For the prediction of the surface roughness of machined workpieces, Artificial Neural Network (ANN) [11,12,13,14], Adaptive Neuro-Fuzzy Inference System (ANFIS) [15,16], Multi-Objective Genetic Algorithm (MOGA) [17], quadratic regression models [18], and other common techniques are used [19]. Mirko Ficko et al. predicted the surface roughness of abrasive water jet machining based on artificial neural network, and used k-fold cross-validation to verify the ANN model, which greatly shortened the time of experimental verification [14]; Jain et al. used the probabilistic simulation method to calculate the number of active abrasives in the AFM process [19]. Liu et al. based their work on the crystal plasticity finite element model and the coupled Eulerian–Lagrangian method to analyze the roughness evolution of constrained surfaces [20]. However, there is currently no research on the application of multiple linear regression to predict surface roughness. The advantages of multiple linear regression are that the calculation is simple, the output is a linear relationship, the model is stable and intuitive, and the correlation between factors can be accurately evaluated.

The novelty of this study is to analyze the influence of different parameters (such as workpiece shape, abrasive flow state, rotational speed and abrasive quality) on the surface roughness of the AFM-processed workpiece, and determine the priority of the effect of machining parameters on the surface roughness. The optimal parameter combination is obtained by designing an orthogonal test, and a prediction model based on the linear regression equation is developed using the orthogonal test data. Finally, the model is tested and analyzed to verify the reliability of the model.

## 2. Materials and Methods

### 2.1. Description of the Test Device

The experiment was conducted by using a self-made gas-solid two-phase flow abrasive pool processing experimental platform, as shown in Figure 1. The processing principle of the self-made abrasive pool processing platform used in this experiment is to mix solid-phase abrasives with gas to obtain gas-solid two-phase flows with various flow states. The abrasive contacts, rubs and collides with the surface of the workpiece, and micro-cuts the surface of the workpiece with the abrasive to realize the finishing of the surface of the workpiece by the gas-solid two-phase flow. Figure 2 is a schematic diagram of the gas-solid two-phase flow abrasive pool finishing process.

Figure 3 shows the cloud diagram of contact force between abrasive particles and workpiece and between abrasive particles simulated by using EDEM software. Figure 3a is the force when the abrasive grains contact the workpiece when no wind force is applied, and the force of the abrasive grains is concentrated on the workpiece. Figure 3b is the force diagram of the abrasive grains when the wind force is applied; it is seen that the force of the abrasive particles on the workpiece gradually decreases. Figure 3c is the force diagram of the abrasive particles after the stable gas-solid two-phase flow is formed. The result shows that the force of the abrasive particles is no longer concentrated around the workpiece, the polishing is more disordered, which can effectively reduce the stress concentration on the surface of the workpiece, prevent the workpiece from rubbing for a long time at a certain point during processing, and generate furrows or pits, so that the processing quality of the workpiece surface is improved.

The workpiece material processed during the experiment was Q235 steel, and the chemical composition of the material is shown in Table 1. The round tubular, square tubular and cylindrical Q235 steel materials were selected for the experiment. The abrasive grains used in the processing of the abrasive pool were white corundum with different particle sizes (as shown in Figure 4). White corundum is suitable for grinding and polishing high carbon steel, high-speed steel and hardened steel, and has excellent grinding and polishing performance. The specific parameters are shown in Table 2.

In the test, five factors, such as workpiece shape, abrasive grain size, gas-solid two-phase flow state, abrasive grain shape, and rotating speed were selected as the main parameters. The parameter ranges are set in Table 3. The minimum air pressure required by the air compressor is 0.735 MPa.

Since there were many factors involved, to reduce the number of experiments without reducing the experimental effect, this paper adopts the orthogonal test to obtain the optimal combination of experiments. The level of each factor was quite different; it was necessary to select the orthogonal experiment method at different levels. Orthogonal experiments at different levels are divided into two methods: the mixed orthogonal experiment method and the quasi-horizontal method. Mixed orthogonal experiments are mostly suitable for situations where there are many levels of a single factor and the levels of other factors are the same, and its expression is as follows:(1)$$Ln\left(m1^{k1}m2^{k2}\right),\;\;n=k1\times \left(m1-1\right)+k2\times \left(m2-1\right)+\cdots kx\times \left(mx-1\right)+1$$
where *L* is an orthogonal table, *n* is the number of experiments, *k* is a factor, and *m* is the number of levels. 

After substituting the data, the number of mixed orthogonal experiments was calculated as 15 times. The quasi-horizontal rule is to complete the factors with fewer levels through repeated levels or other methods to ensure that each factor can meet the same level, and then design a standard orthogonal experiment. After the orthogonal table was quasi-horizontal transformation, generally the table that has been transformed into a quasi-horizontal column is listed as a quasi-horizontal column. According to Table 3, after the shape of the abrasive grains was filled by the pseudo-horizontal method, the standard orthogonal experiment table of four factors and three levels L9 (43) can be obtained. The orthogonal experiment table is shown in Table 4.

The gas-solid two-phase flow state, workpiece shape and abrasive particle shape are all dimensionless factors. To simplify the orthogonal experiment, the above three parameters were digitized, and the gas-solid two-phase flow state can be replaced by the Reynolds number; the value range was set to 2000, 4000, and 6000. The shape of the workpiece can be replaced by the contact surface between the workpiece and the abrasive flow; the contact between the workpiece and the abrasive flow is generally the entire curved surface, the contact of the square tube is four planes, and the contact of the cylinder is the bottom plane. Process these three values into data and use 1/3, 4/3, and 1/2 instead. There were two shapes of abrasives, spherical and irregular, and numbers 1 and 2 can be used to represent spherical and irregular shapes. For testing the overall effect of the abrasive pool processing, it was necessary to ensure the continuity of the experiment. Therefore, the same workpiece was divided into three stages for experimental testing, that is, the orthogonal experiment of the workpiece under 24, 80 and 120 mesh abrasives, respectively. The roughness of the workpiece will be measured in each stage of the experiment, and the experimental workpiece will not be replaced during the period, but the experimental measurement was still required for each stage of processing. After the experiment, the surface roughness of the workpiece was measured with a roughness meter, and the measurement points are shown in Figure 5.

### 2.2. Analysis of Orthogonal Experiment Results

Firstly, the surface roughness of each marked point of the round tube workpiece under 24-mesh abrasive was analyzed by the intuitive analysis method. Figure 6 shows the average of five measurement roughness values of the same marked points of the round tube under the orthogonal experimental conditions.

To fully analyze the polishing effect of the workpiece, it is necessary to analyze the results of each point. Take the analysis of point 1 as an example. Table 5 shows the average and range of surface roughness changes at point 1. K represents the sum of surface roughness changes measured by different factors at the same level, K_1_–K_3_ are rust removal experiments, K_4_–K_6_ are rough polishing experiments, and K_7_–K_9_ are fine polishing experiments. k represents the average value of different factors at the same level. R represents the extreme difference, R = K_max_ − K_min_, the larger the extreme difference, the greater the influence of this factor on finishing. A, B, C and D are used to replace the spindle speed, gas-solid two-phase flow, workpiece shape and abrasive particle shape in the processing factors, respectively. The subscripts 1, 2, and 3 are the level 1, level 2, and level 3 corresponding to the factor, respectively.

It is seen in Table 5 that the range of the D factor is the smallest when the abrasive pool is processed, indicating that the shape of the abrasive grains has the smallest influence on the surface roughness; the range of C is the largest, indicating that the shape of the workpiece has the greatest influence on the surface roughness of the workpiece. With the reduction in the abrasive and the improvement of the polishing quality, the influence of the gas-solid two-phase flow on the workpiece gradually increases, and the range value gradually increases from 0.401 μm at the beginning to 0.9397 μm at the end. With the improvement of the processing quality, the influence of the particle shape of the abrasive on the workpiece gradually increases. From the range, it is seen that the four factors act on the workpiece at the same time, the optimal combination of roughness is always A_3_B_2_C_1_D_2_, that is, the spindle speed is 1200 rpm, the gas-solid two-phase flow state is turbulent, the shape of the workpiece is a circular tube, and the shape of the abrasive particles is irregular, which shows that the workpiece shape has the greatest influence on the machining effect until the final polishing experiment.

However, it is impossible to quantitatively judge which factor has the greatest impact on the surface processing effect of the workpiece at which level by only using the average value and the range. To more intuitively determine which factor has a comparatively large influence on the roughness, it is necessary to make a factor trend graph for the orthogonal result. Continue to take point 1 as an example, and evaluate it according to the average value of roughness. Under the same factor, which type of surface roughness is the smallest, indicating that this level is the optimal level. The abscissa represents the different levels of different factors, the ordinate represents the surface roughness value during processing, and the final trend graph of the factors is formed, as shown in Figure 7.

It is presented in Figure 7 that the trend of the C factor in the same experimental stage changes greatly, indicating that the shape of the workpiece has the greatest influence on the processing of the abrasive pool. With the gradual precision of the experimental processing and the decrease in the number of abrasive grains, the influence of the gas-solid two-phase flow on the processing effect was more and more obvious. The influence of the abrasive shape on the roughness of the machined workpiece was gradually increasing. The slope of factors B_1_-B_3_ gradually increases, indicating that 120-mesh abrasive grains may still be relatively large abrasive grains for abrasive pool processing, and were not the most suitable abrasive grains for processing. It can be speculated that the most suitable abrasive pool processing may be a more precise finishing. It can be seen from Figure 7 that the optimal combination of the rust removal experiment at point 1 is A_3_B_1_C_1_D_2_, the optimal combination of the rough polishing experiment is A_2_B_2_C_1_D_3_, and the optimal combination of the fine polishing experiment is A_1_B_2_C_1_D_2_.

By analogy, the three-stage experiments of the remaining detection points were comprehensively analyzed. Table 6 shows the influence degree of each factor in all points and the optimal parameter combination table [11].

The analysis of the results in Table 6 shows that the more factors and levels in the optimal combination, the greater the impact on the surface machining quality of the workpiece:

As the first influencing factor, A appears 0 times, B is 0 times, C is 15 times, and D is 0 times; as the second influencing factor, A appeared 3 times, B was 11 times, C was 0 times, and D was 1 time; as the third influencing factor, A appeared 10 times, B was 3 times, C was 0 times, and D was 2 times; as the fourth influencing factor, A appears 2 times, B is 1 time, C is 0 times, and D is 12 times.

From the results of all stages, the primary and secondary degrees of each factor affecting the workpiece are workpiece shape, abrasive flow state, rotating speed, and abrasive shape. In the same way, under the same factor, the one with the highest number of levels is the optimal condition, and the results are specified in Table 7.

Therefore, the comprehensive optimal parameter combination for abrasive pool processing is A_3_B_2_C_1_D_2_.

## 3. Roughness Prediction Model

The optimal parameter combination within a certain range was obtained through the orthogonal experiment, and a surface roughness model was established on this basis to predict the surface roughness of the workpiece, reduce the number of experiments and save the experiment time. Based on the orthogonal experimental data, the multiple linear regression analysis methods were used to establish the surface roughness prediction model of abrasive pool machining as the basis of the surface roughness prediction of abrasive pool machining.

Prediction requirements for the surface roughness of abrasive pool processing: while the surface quality of the workpiece meets the requirements, the processing efficiency was maximized as much as possible. The processing parameters of the experiment, namely, spindle speed, gas-solid two-phase flow state, workpiece shape, and abrasive shape are used as the input end, and the surface roughness of the workpiece was used as the output end, and the results should be more accurate, which can provide convenience for the subsequent experimental progress. It can provide an important reference for subsequent abrasive pool processing.

### 3.1. Multiple Linear Regression Prediction Models

Suppose there is a linear correlation between the random variable *y* and the *p* independent variables *x*_1_, *x*_2_, …, *x_p_*, the actual sample size is *n*, and the observed value is *x_i_*_1_, *x_i_*_2_, …, *x_ip_*, *y_i_* (*i* = 1, 2,…, *n*). Then, the *n* observations can be expressed as Equation (2):(2)$$\left\{\begin{array}{c} y_{1}=\beta_{0}+\beta_{1}x_{11}+\beta_{2}x_{12}+\cdots +\beta_{p}x_{1p}+\varepsilon_{1}\\ y_{2}=\beta_{0}+\beta_{1}x_{21}+\beta_{2}x_{22}+\cdots +\beta_{p}x_{2p}+\varepsilon_{2}\\ \cdots \cdots \\ y_{n}=\beta_{0}+\beta_{1}x_{n1}+\beta_{2}x_{n2}+\cdots +\beta_{p}x_{np}+\varepsilon_{n} \end{array}\right.$$

Among them, *β*_0_, *β*_1_, …, *βn* are position parameters, *x*_1_, *x*_2_, …, *x_p_* are *p* general variables that can be accurately measured and controlled, and *ε*_1_, *ε*_2_, …, *ε_p_* are random errors. It is assumed that *ε_i_* are independent random variables that obey the same normal distribution N(0, σ). Equation (3) can be represented by a matrix:(3)y=Xβ+ε
where
(4)$$y=\left(\begin{array}{c} y_{1}\\ y_{2}\\ \vdots \\ y_{n} \end{array}\right)\;X=\left(\begin{array}{ccccc} 1 & x_{11} & x_{11} & \ldots & x_{11}\\ 1 & x_{11} & x_{11} & \ldots & x_{11}\\ 1 & \vdots & \vdots & \ldots & \vdots \\ 1 & x_{11} & x_{11} & \ldots & x_{11} \end{array}\right)\;\beta =\left(\begin{array}{c} \beta_{1}\\ \beta_{2}\\ \vdots \\ \beta_{n} \end{array}\right)\;\;\varepsilon =\left(\begin{array}{c} \varepsilon_{1}\\ \varepsilon_{2}\\ \vdots \\ \varepsilon_{n} \end{array}\right)$$

Solving the multiple linear regression equation is to establish the multiple linear regression equation by solving the estimated value *b* of *β*:(5)$$\hat{y}=b_{0}+b_{1}x_{1}+b_{2}x_{2}+\cdots +b_{p}$$

There is a linear relationship between milling and the surface roughness of the workpiece. Through the similarity principle, it can be inferred that the abrasive pool machining also has this relationship with the workpiece. Tipnis et al. established an empirical model of surface roughness for cutting speed, feed and depth of cut. Based on this, scholars established a general model of milling and surface roughness:(6)$$Ra=ca_{p}^{b1}a_{e}^{b2}n^{b3}f_{z}^{b4}$$
where *c* is the correction coefficient of the workpiece material, *a_p_* is the cutting depth, *a_e_* is the cutting width, *f* is the feed amount, and *b_1_*, *b_2_*, *b_3_*, and *b_4_* are to be estimated. By the second law of similarity, for similar physical quantities, the similarity criterion should be the same. After substituting the abrasive pool processing parameters into the above equation, we can get:(7)$$Ra=cm^{b1}Re_{pmf}{}^{b2}n^{b3}M^{b4}$$
where *m* represents the shape of the workpiece, *Re_pmf_* is the Reynolds number of the abrasive in the abrasive pool, that is, the flow state after the gas-solid two-phase flow is formed, *M* is the shape of the abrasive, and *n* is the spindle speed. Since the output value is a linear function, it is necessary to linearize the nonlinear function of the above equation, take the logarithm of both sides, and Equation (8) can be obtained:(8)$$lgRa=lgc+b1lgm+b_{2}lgRe_{pmf}+b_{3}lgn+b_{4}lgM$$

Suppose that:(9)$$lgRa=y,lgc=b_{0},lgm=x_{1},lgRe_{pmf}=x_{2},lgn=x_{3},lgM=x_{4}$$
transform it into:(10)$$y=b_{0}+b_{1}x_{1}+b_{2}x_{2}+b_{3}x_{3}+b_{4}x_{4}$$

In Equation (10), *y* is a statistical variable. The regression coefficient *b* can be obtained using Equation (11):(11)$$b={\left(X^{\prime }X\right)}^{-1}X^{\prime }Y$$

First, solve the model. The least-squares method was used to find *b*, and the surface roughness prediction model of the rust removal stage was first calculated. To optimize the data, the surface roughness measurements were performed on multiple points of different workpieces in the previous test, each factor level corresponds to five roughness results. The points with the same factor and level were taken as the average surface roughness. It can be obtained by calculation:
(12)$$X=\left\{\begin{array}{cccc} 2.778 & 3.301 & -0.476 & 0\\ 2.778 & 3.602 & 0.125 & 0.301\\ 2.778 & 3.778 & -0.301 & 0.301\\ 2.954 & 3.301 & 0.125 & 0.301\\ 2.954 & 3.602 & -0.301 & 0\\ 2.954 & 3.778 & -0.476 & 0.301\\ 3.079 & 3.301 & -0.301 & 0.301\\ 3.079 & 3.602 & -0.476 & 0.301\\ 3.079 & 3.778 & 0.125 & 0 \end{array}\right.Y=\left\{\begin{array}{c} 0.6784\\ 0.7118\\ 0.7436\\ 0.7218\\ 0.7023\\ 0.6890\\ 0.6958\\ 0.6627\\ 0.7197 \end{array}\right.$$

The multiple linear regression equation can be solved by using the regress command of Matlab. The regression coefficient *b*, residual *r*, and *F* value can be obtained by the code:[*b*, bint, *r*, rint, stats] = regress (*Y*, *X*).
(13)$$b={\left[\;\begin{array}{ccccc} 1.6487 & -0.0265 & 0.0208 & 0.0562 & -0.0083 \end{array}\right]}^{T}$$

Then, conduct residual analysis. Randomness and unpredictability are key components of all regression models, and random error needs to be random and unpredictable. After obtaining the regression model, it is necessary to ensure that the residuals are untraceable, usually based on residual plots to analyze the reliability of the data and whether the model is correct.

As can be seen from Figure 8, the residual distribution of the model is close to the normal distribution and belongs to the standardized residual. It shows that the model is not disturbed by specific factors, which ensures the correctness of the model. Therefore, the surface roughness prediction model in the rust removal stage is:(14)$$Ra=e^{1.7071}m^{-0.0348}Re_{pmf}{}^{0.0230}n^{0.0898}M^{0.0146}$$

In the same way, the residual analysis of the surface roughness prediction model in the rough polishing stage and the fine polishing stage was conducted. The residual distribution diagram of the rough throwing stage is shown in Figure 9.

As can be seen from Figure 9 the residual distribution in the rough throwing stage is a normal distribution, so the roughness prediction model in the rough throwing stage is:(15)$$Ra=e^{0.0599}m^{0.0275}Re_{pmf}{}^{0.0424}n^{0.1429}M^{-0.0074}$$

As can be seen from Figure 10, the residual distribution is a normal distribution, so the roughness prediction model in the fine polishing stage is:(16)$$Ra=e^{-1.3465}m^{0.0808}Re_{pmf}{}^{0.1019}n^{0.6812}M^{-0.0597}$$
(17)$$y=\beta_{0}+\beta_{1}x_{1}+\beta_{2}x_{2}+\cdots +\beta_{p}x_{p}$$

### 3.2. Significance Test of Prediction Model

Firstly, use the *F-test* to test the overall significance level of the model. The significance test for the multiple linear regression equation is to test whether the parameters in Equation (18) are significantly different from 0.
(18)$$y=b_{0}+b_{1}x_{1}+b_{2}x_{2}+b_{3}x_{3}+\cdots +b_{n}x_{n}$$
suppose that:(19)$$\left\{\begin{array}{c} H0:b1=b2=\cdots =bk=0\;F\leq F_{\alpha }\\ H1:bj\left(j=1,2,\ldots ,k\right)\;not\;all\;0\;F>F_{\alpha } \end{array}\right.$$

Since *y* obeys the normal distribution, the sum of the squares of the samples corresponding to *y* still obeys the square of the normal distribution, so:(20)$$ESS=\overset{n}{\underset{1}{\sum }}{\left({\hat{y}}_{t}-\bar{y}\right)}^{2}/\sigma^{2}\sim x^{2}\left(k\right)$$
(21)$$RSS=\overset{n}{\underset{1}{\sum }}{\left(\bar{y}-{\hat{y}}_{t}\right)}^{2}/\sigma^{2}\sim x^{2}\left(n-k-1\right)$$

The equation for the *F-test* is:(22)$$F=\frac{ESS/k}{RSS/\left(n-k-1\right)}$$

Among them, *RSS* is derived from the regression sum of squares, and *ESS* is derived from the residual sum of squares, both of them follow the square of the normal distribution, *n* represents the amount of data, and *k* is the number of calculation parameters. Calculated by Matlab, the *F* value of the rust removal stage is 15.352 greater than 15.22 [*F* (4,5) (α = 0.01)], that is, the assumption H1 is accepted, which means that *b_j_* (*j* = 1, 2,…, *k*) is not all 0, indicating that the prediction model of the rust removal stage is significant.

*T-test* uses t-distribution theory to infer the probability of differences, that is, to test the significance of the independent variables in the equation. Provided the independent variables in each group show a normal distribution, it means that there are consistent differences between groups, that is, the rejection H0 can be maximized. The Equation *T-test* is:(23)$$T=\frac{\bar{X}-\mu }{\frac{\sigma_{X}}{\sqrt{N}}}$$
where $\bar{X}$ is the mean value of each factor, *N* is the number of samples, and *T* needs to follow a normal distribution. The significance test is conducted between the factors. If *T* < *Tα*, hypothesis H0 is accepted, indicating that the two data are similar and there is no discrimination; if *T* > *Tα*, hypothesis H1 is accepted, indicating that the two dates are from different distribution data, and differentiated. Check the *T* value distribution table, when *α* = 0.01, *Tα* is 2.8214. After substituting the orthogonal experimental data into Equation (23), the *T* value of each coefficient is obtained as shown in Table 8. It can be seen that the absolute value of *T* of each factor is greater than *Tα*, so the assumption H0 is accepted, and there is a clear distinction between independent variables. It shows that each factor in the prediction model has a significant effect on the surface roughness.

Similarly, the *F* value of the rough throwing stage is 32.156 greater than 15.22 [*F* (4,5) (α = 0.01)], the absolute value of *T* is greater than *Tα* (2.8214), the *F* value of the fine throwing stage is 50.317 greater than 15.22 [*F* (4,5) (α = 0.01)], and the absolute value of *T* is greater than *Tα* (2.8214).

In summary, the surface roughness models of each polishing stage have good significance, high reliability and data fit.

### 3.3. Significance Test of the Prediction Model

Abrasive pool processing is most effective in fine polishing. To verify the accuracy of the surface roughness prediction model, 120-mesh abrasive particles were selected to conduct the precision polishing experiment on the cylindrical workpiece. Five combinations were randomly selected for the experiment. The feature points were measured, and the above experimental conditions were substituted into the prediction model. The experimental results and the predicted results are shown in Table 9. Figure 11 illustrated a comparison diagram of the predicted value and the measured value:

The average error ratio calculation equation is:(24)$$\delta =\frac{1}{n}\overset{n}{\underset{1}{\sum }}\left|\frac{M_{m}-M_{f}}{M_{m}}\right|\times 100\%$$
where *M_m_* is the measured value and *M_f_* is the predicted value. It can be seen from Table 10 that the maximum error between the predicted value and the actual value of the surface roughness processed by the abrasive pool is 0.068 μm, the minimum error is 0.0094 μm, the error ratio is basically below 10%, and the average error is only 6.44%, indicating that in the fine polishing experiment stage, the roughness prediction model is basically consistent with the actual value, and the established surface roughness prediction model can predict the surface roughness of the abrasive pool well.

The roughness prediction model of the rust removal and rough polishing experiments is the same as the roughness prediction model of the fine polishing experiment. The results are shown in Table 11. Experiments 1–5 are rust removal experiments, and 6–10 are rough polishing experiments. R_1_ is the mean value of the rust removal stage, and R2 is the mean value of the rough throwing stage.

From the perspective of error ratio, the prediction model of workpiece surface roughness during rust removal is more accurate, with an error ratio of only 1.4%, followed by the fine polishing experiment with an error ratio of 6.44%, and the last rough polishing experiment with an error ratio of 6.8%. Observing the error value, it can be found that the error value during fine polishing is only 0.00844 μm. From the point of view of the error value, the average error of the surface roughness prediction model during fine polishing is the smallest, and the surface roughness model is more accurate. In general, the errors of the surface roughness prediction models for rust removal, rough polishing and fine polishing are all within an appropriate range, and these three surface roughness models can predict the surface roughness of the actual abrasive pool machining.

## 4. Conclusions

The influence of the parameters involved in the abrasive pool machining on the workpiece machining quality was evaluated by designing an orthogonal test, and the optimal parameter combination was determined by using the range method and factor trend diagram. The surface roughness prediction model was established in three stages, and the accuracy of the model was verified. The specific conclusions are as follows:(1)The results of the orthogonal test show that the abrasive pool machining has the advantages of high machining efficiency and small surface loss, and the optimal machining parameter combination of A_3_B_2_C_1_D_2_ was obtained.(2)The surface roughness prediction model of abrasive pool machining was obtained through multiple linear regression equations, residual analysis and significance analysis were carried out on the model, and MATLAB was used for solution analysis to determine the randomness and the randomness of the surface roughness prediction model.(3)The actual surface roughness was obtained through the experiment and compared with the predicted value of the model. As a result, the maximum value of the average error between the predicted value of the surface model and the actual value was 0.339 μm, and the minimum value was 0.008 μm. Predict surface roughness for abrasive pool machining.

In future research, the influence of other process parameters on the grinding and polishing accuracy should be discussed in more detail, and the flow regime of the abrasive pool should be systematically analyzed through hydrodynamic simulation to better predict the surface roughness of the machined workpiece.

## Figures and Tables

**Figure 1 micromachines-13-01649-f001:**
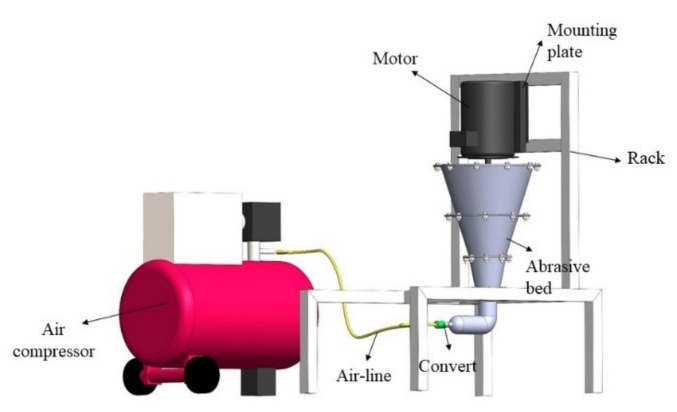
Three-dimensional diagram of abrasive pool test bench.

**Figure 2 micromachines-13-01649-f002:**
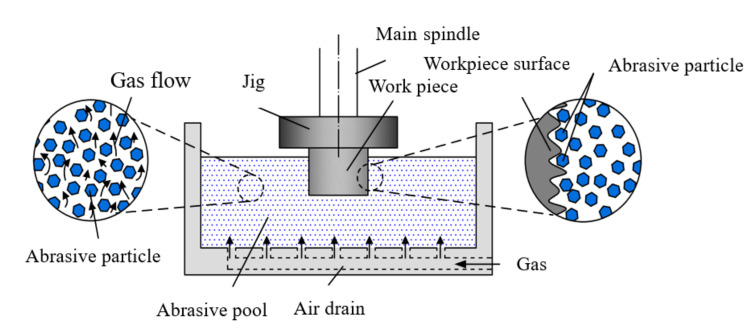
The principle of gas-solid two-phase flow finishing.

**Figure 3 micromachines-13-01649-f003:**
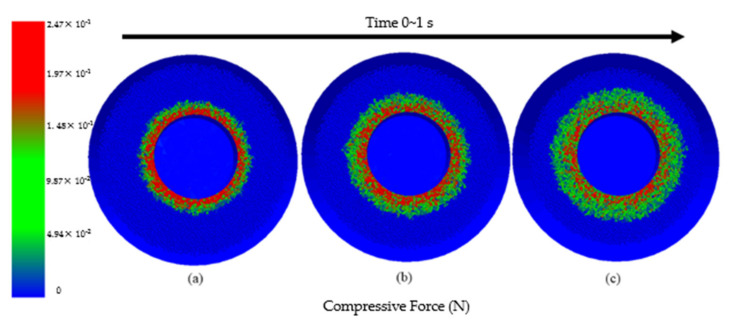
Abrasive contact with workpiece diagram. (**a**); not applied wind force (**b**) applied wind power; (**c**) stable gas-solid two-phase flow is formed.

**Figure 4 micromachines-13-01649-f004:**
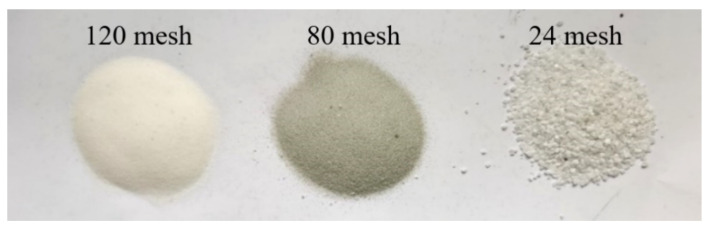
Abrasive.

**Figure 5 micromachines-13-01649-f005:**
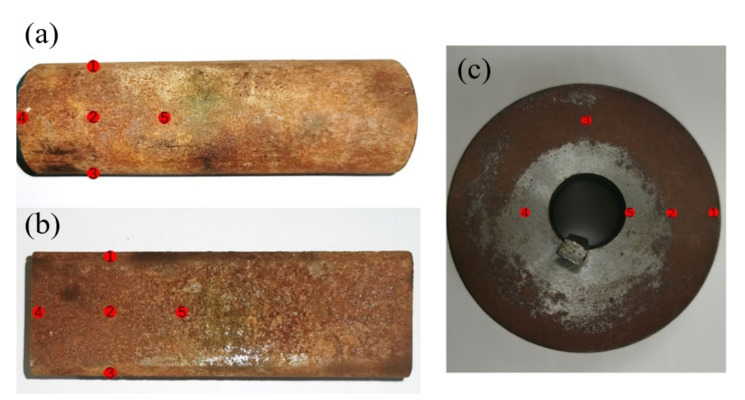
Workpiece diagram: (**a**) round tube; (**b**) square tube; (**c**) cylinder.

**Figure 6 micromachines-13-01649-f006:**
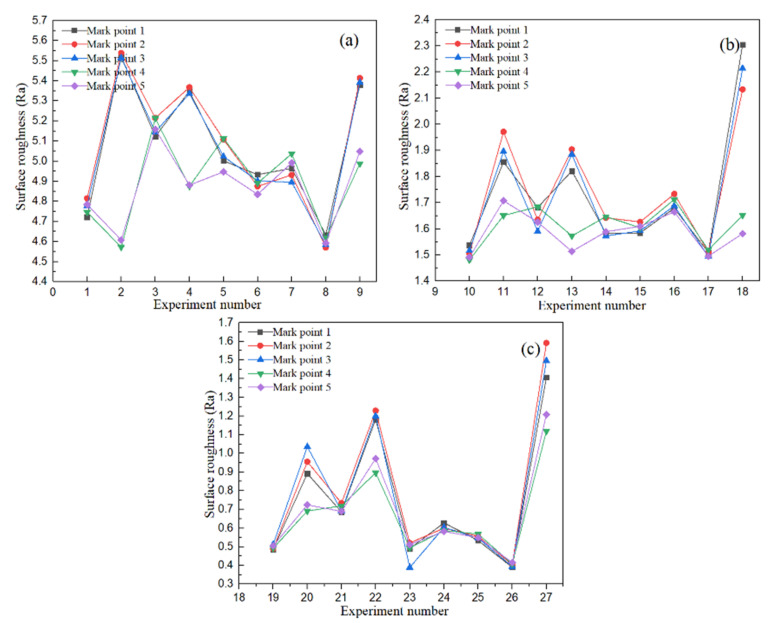
The roughness height of circular tube marking point by orthogonal experiment: (**a**) 120 mesh abrasive; (**b**) 80 mesh abrasive; (**c**) 24 mesh abrasive.

**Figure 7 micromachines-13-01649-f007:**
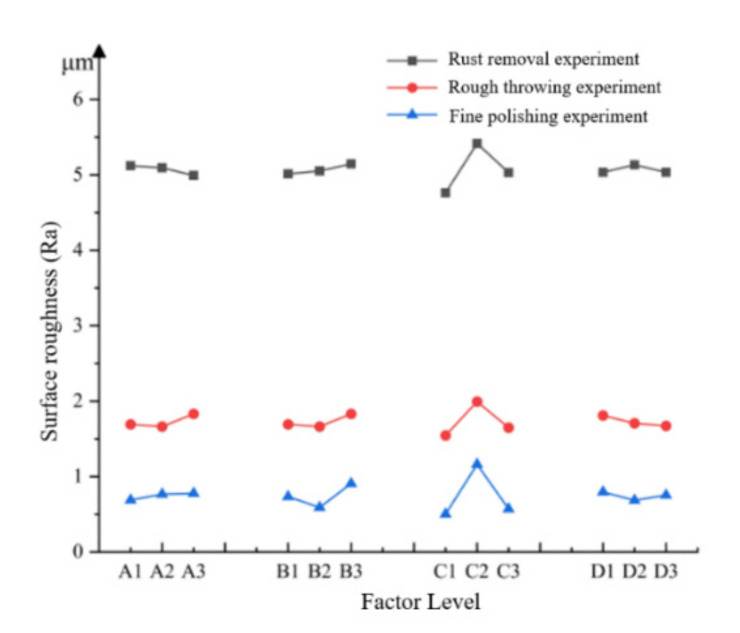
Factor trend of point 1 by test.

**Figure 8 micromachines-13-01649-f008:**
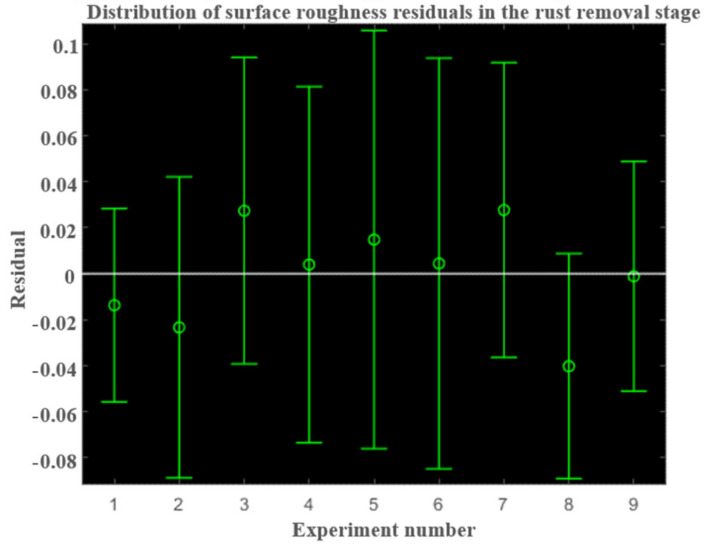
Distribution of surface roughness residuals in the rust removal stage.

**Figure 9 micromachines-13-01649-f009:**
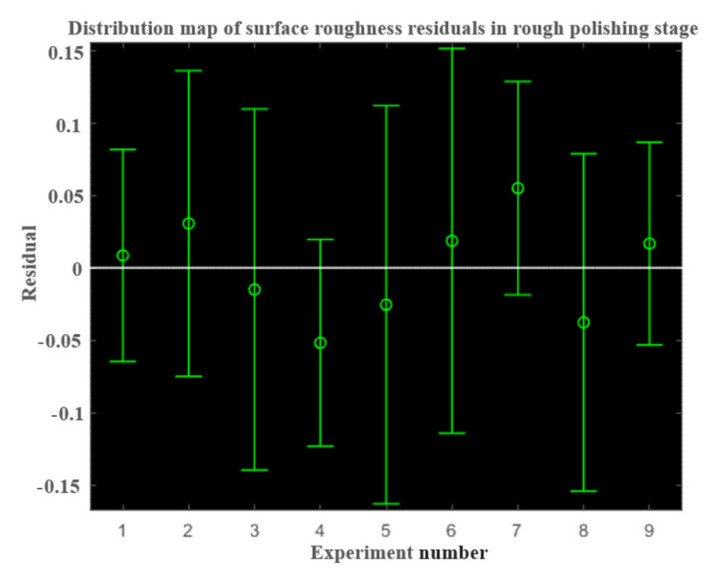
Distribution of surface roughness residuals in the rough polishing stage.

**Figure 10 micromachines-13-01649-f010:**
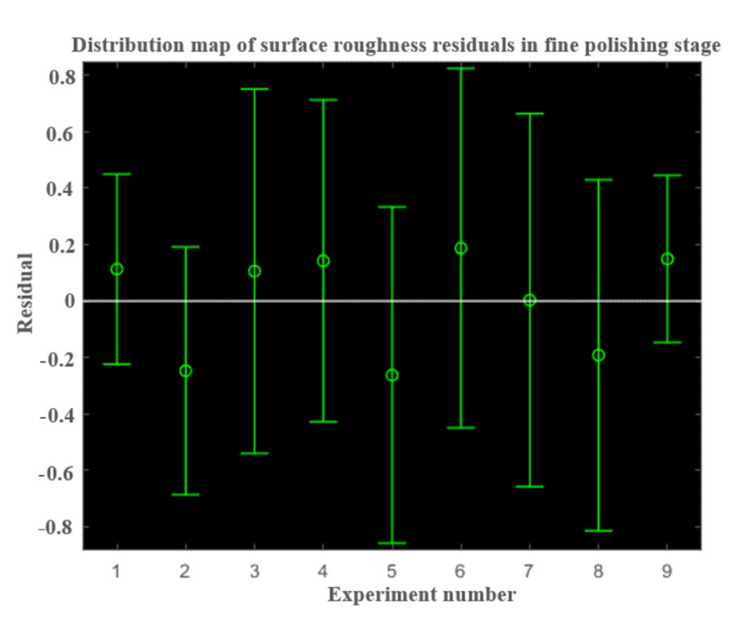
The residual distribution of the fine throwing stage.

**Figure 11 micromachines-13-01649-f011:**
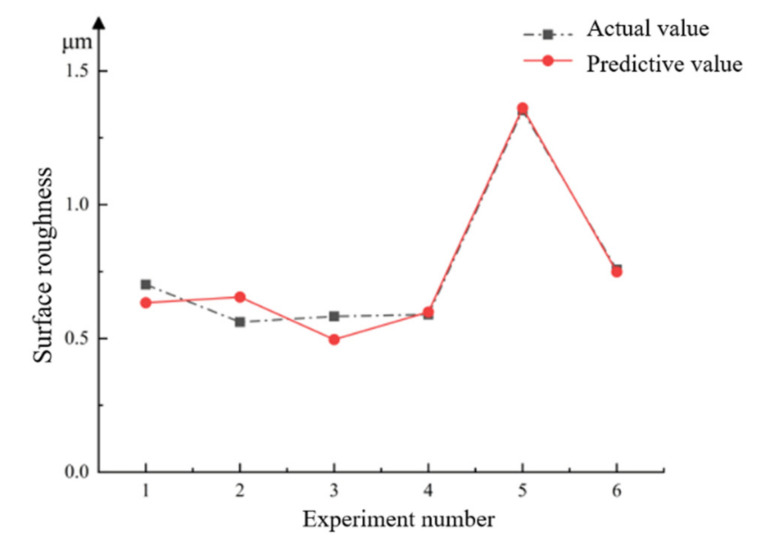
The measured and predicted values of the finishing polish.

**Table 1 micromachines-13-01649-t001:** Workpiece material properties.

Material	Yield Strength /MPa	Hardness /HB	Melting Point/°C	C/%	Mn/%	Si/%	S/%
Q235	235	165	1600	0.12–0.20	0.3–0.65	0.3	0.05

**Table 2 micromachines-13-01649-t002:** Physical properties of white corundum.

Material	Hardness	Shape	Density (Bulk) g/cm^3^
Al_2_O_3_	9.0	spherical/irregular	1.53–1.99

**Table 3 micromachines-13-01649-t003:** Orthogonal experiment parameter range setting.

	Processing Parameters			
	Workpiece Shape	Gas-solid Two-Phase Flow	Abrasive Shape	Spindle Speed/rpm
1	round tube	bulk fluidized bed	spherical	600
2	square tube	turbulent fluidized bed	irregular	900
3	cylinder	spouted fluidized bed	/	1200

**Table 4 micromachines-13-01649-t004:** Orthogonal experiment table.

Experiment Number	Factors			
	Spindle Speed/rpm	Gas-Solid Two-Phase Flow	Workpiece Shape	Abrasive Shape
1	600	2000	1/3	1
2	600	4000	4/3	2
3	600	6000	1/2	2
4	900	2000	4/3	2
5	900	4000	1/2	1
6	900	6000	1/3	2
7	1200	2000	1/2	2
8	1200	4000	1/3	2
9	1200	6000	4/3	1

**Table 5 micromachines-13-01649-t005:** Range analysis results at point 1.

	Spindle Speed (A)	Gas-Solid Two-Phase Flow (B)	Workpiece Shape (C)	Abrasive Shape(D)
**K_1_**	15.336	15.036	14.288	15.105
**K_2_**	15.248	15.156	16.251	15.419
**K_3_**	14.979	15.437	15.09	15.105
**K_4_**	5.0754	5.0377	4.6318	5.4249
**K_5_**	4.9896	4.9491	5.9825	5.1204
**K_6_**	5.4938	5.572	4.9445	5.0135
**K_7_**	2.0646	2.2013	1.4946	2.3848
**K_8_**	2.2904	1.7729	3.482	2.0438
**K_9_**	2.332	2.7128	1.7104	2.2584
**k_1_**	5.122	5.012	4.763	5.035
**k_2_**	5.095	5.052	5.417	5.140
**k_3_**	4.933	5.146	5.03	5.035
**k_4_**	1.6918	1.6792	1.5439	1.8083
**k_5_**	1.6632	1.6497	1.9941	1.7068
**k_6_**	1.8312	1.8573	1.6481	1.6711
**k_7_**	0.6882	0.7337	0.4982	0.7949
**k_8_**	0.7634	0.5909	1.1606	0.6812
**k_9_**	0.7773	0.9042	0.5701	0.7528
**R_1_**	0.387	0.401	1.963	0.314
**R_2_**	0.4184	0.6229	1.3507	0.3045
**R_3_**	0.2672	0.9397	1.9872	0.3408
**The biggest influencing factor in the rust removal stage**	C	B	A	D
**The biggest influencing factor in rough throwing stage**	C	B	A	D
**The biggest influencing factor in the polishing stage**	C	B	D	A
	Rust removal experiment	Rough throwing experiment	Fine polishing experiment	/
**The best combination of processing quality**	A_3_B_2_C_1_D_2_	A_3_B_2_C_1_D_2_	A_3_B_2_C_1_D_2_	/

**Table 6 micromachines-13-01649-t006:** The influence degree of each factor and the optimal parameter combination table of each experiment.

Marked Point		1	2	3	4	5
**Rust removal experiment**	**Influence level**	CBAD	CABD	CABD	CBAD	CBAD
	**Optimal** **combination**	A_3_B_1_C_1_D_2_	A_3_B_2_C_1_D_2_	A_1_B_1_C_1_D_1_	A_1_B_2_C_2_D_2_	A_3_B_2_C_1_D_2_
**Rough** **throwing** **experiment**	**Influence level**	CBAD	CDBA	CADB	CBAD	CBAD
	**Optimal** **combination**	A_2_B_2_C_1_D_3_	A_1_B_1_C_1_D_1_	A_3_B_2_C_1_D_2_	A_1_B_1_C_1_D_1_	A_1_B_1_C_1_D_1_
**Fine polishing experiment**	**Influence level**	CBAD	CBAD	CBDA	CBAD	CBAD
	**Optimal** **combination**	A_1_B_2_C_1_D_2_	A_3_B_2_C_1_D_2_	A_2_B_2_C_2_D_1_	A_3_B_2_C_1_D_2_	A_3_B_2_C_1_D_2_

**Table 7 micromachines-13-01649-t007:** Optimal level.

Factors	Level 1	Level 2	Level 3
**A**	5	1	9
**B**	4	11	0
**C**	13	2	0
**D**	5	10	0

**Table 8 micromachines-13-01649-t008:** Distribution of T value in each stage of abrasive pool processing.

Regression Coefficients	b_1_	b_2_	b_3_	b_4_
**Derusting stage T value**	3.42	4.28	3.46	6.84
**Rough throwing stage T value**	5.68	−3.51	9.67	−4.85

**Table 9 micromachines-13-01649-t009:** Parameter settings of surface roughness prediction experiment.

Experiment Number	Experimental Parameters			
	A	B	C	D
**1**	600	6000	1/2	2
**2**	900	4000	1/2	1
**3**	900	6000	1/3	2
**4**	1200	2000	1/2	2
**5**	1200	6000	4/3	1

**Table 10 micromachines-13-01649-t010:** The measured and predicted values of the finishing polish.

Experiment	Measured Value	Predictive Value	Difference	Error Ratio
**1**	0.7013	0.6333	0.068	9%
**2**	0.5912	0.6545	−0.0633	10%
**3**	0.5525	0.4961	0.0664	11%
**4**	0.5895	0.5989	−0.0094	1.5%
**5**	1.3521	1.3616	−0.0095	0.7%
**R**	0.75732	0.74888	0.00844	6.44%

**Table 11 micromachines-13-01649-t011:** The measured and predicted values of rust removal and rough polishing.

Experiment	Measured Value	Predictive Value	Difference	Error Ratio
**1**	5.135	5.063	0.072	1.4%
**2**	5.036	4.967	0.069	1.3%
**3**	4.87	4.8	0.07	1.4%
**4**	4.955	4.833	0.122	2.4%
**5**	5.365	5.256	0.109	2%
**6**	1.645	1.658	0.013	0.7%
**7**	1.987	1.648	0.339	17%
**8**	1.612	1.574	0.038	2.3%
**9**	1.693	1.605	0.088	2.1%
**10**	2.115	1.944	0.171	8.0%
**R_1_**	5.0736	4.9978	−0.0758	1.4%
**R_2_**	1.8104	1.6858	−0.1246	6.8%

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
