# Peer review of "Prediction of Surface Roughness in Gas-Solid Two-Phase Abrasive Flow Machining Based on Multivariate Linear Equation"

_micromachines, 2022, doi:10.3390/mi13101649_

Round 1

Reviewer 1 Report

1.The parameters of the processing experiment in the paper are missing, such as abrasive concentration, air pressure, etc.   2.Line 16, The results show that the average error between  the predicted value of the surface roughness model and the actual value is 0.339 μm and the minimum value is 0.008 μm.The 0.339 μm that appears here is the average error value, but this data in the paper is only the error value of one result, and the data of 0.008 μm does not see the source in the paper chart. Please explain if these data are used correctly   3.Line 400,Table 10 that the maximum error between the predicted value and the actual value of the  surface roughness processed by the abrasive pool is 0.0933 μm, the minimum error is  0.0094 μm, but in Table 10, the data of 0.0933μm does not appear, Please explain if these data are used correctly   4.Line 414, Observing the error value, it can be found that the error value during fine polishing is only 0.008 μm, but in Table 11, the data of 0.008μm does not appear. Please explain if these data are used correctly.   5.The layout of the paper is not standardized,

Author Response

Dear Editor:

The authors would like to thank the editor for providing valuable comments and suggestions on this manuscript. Our responses to the comments and suggestions are as follows and the manuscript was revised accordingly with the revised text highlighted in yellow.

Comment 1: The parameters of the processing experiment in the paper are missing, such as abrasive concentration, air pressure, etc.

Response: The initial experiments are planned to use abrasive grains of 24, 80, and 120 mesh; the minimum air pressure required by the air compressor is 0.735 MPa (Line 116-117).

Comment 2: Line 16, The results show that the average error between the predicted value of the surface roughness model and the actual value is 0.339 μm and the minimum value is 0.008 μm.The 0.339 μm that appears here is the average error value, but this data in the paper is only the error value of one result, and the data of 0.008 μm does not see the source in the paper chart. Please explain if these data are used correctly.

Response: The "average error value of 0.339" in the abstract is a writing error, and 0.339 should be the maximum error value here; the average error value of 0.008 is actually 0.00844 in Table 10, which has been revised (Line 15-16).

Table 10. The measured and predicted values of the finishing polish.

Experiment

Measured value

Predictive value

Difference

Error ratio

1

0.7013

0.6333

0.068

9%

2

0.5912

0.6545

-0.0633

10%

3

0.5525

0.4961

0.0664

11%

4

0.5895

0.5989

-0.0094

1.5%

5

1.3521

1.3616

-0.0095

0.7%

R

0.75732

0.74888

0.00844

6.44%

Table 11. The measured and predicted values of rust removal and rough polishing.

Experiment

Measured value

Predictive value

Difference

Error ratio

1

5.135

5.063

0.072

1.4%

2

5.036

4.967

0.069

1.3%

3

4.87

4.8

0.07

1.4%

4

4.955

4.833

0.122

2.4%

5

5.365

5.256

0.109

2%

6

1.645

1.658

0.013

0.7%

7

1.987

1.648

0.339

17%

8

1.612

1.574

0.038

2.3%

9

1.693

1.605

0.088

2.1%

10

2.115

1.944

0.171

8.0%

R1

5.0736

4.9978

-0.0758

1.4%

R2

1.8104

1.6858

-0.1246

6.8%

Comment 3: Line 400,Table 10 that the maximum error between the predicted value and the actual value of the  surface roughness processed by the abrasive pool is 0.0933 μm, the minimum error is  0.0094 μm, but in Table 10, the data of 0.0933μm does not appear, Please explain if these data are used correctly.

Response: The data was misused, the maximum error between the predicted and actual values of the surface roughness for the abrasive pool was 0.068 μm and the minimum error was 0.0094 μm, revised (Line 393).

Comment 4: Line 414, Observing the error value, it can be found that the error value during fine polishing is only 0.008 μm, but in Table 11, the data of 0.008μm does not appear. Please explain if these data are used correctly.

Response: Was shown in Table 10, 0.008μm is an approximation, the actual value is 0.00844 μm

Comment 5: The layout of the paper is not standardized.

Response: Revised

Reviewer 2 Report

The present work explored the surface roughness prediction model in Gas-Solid Two-Phase Abrasive Flow Machining. Besides, by means of orthogonal test method, the optimal process parameters combination was obtained though analyzing the relationship between the parameters with surface roughness values. While, through carefully reviewing, it is suggested that this manuscript should be major revised before accepting. Detailed comments are listed in below.

1. The sentences in present articles needs to be carefully polished. Besides, the Abstract and Introduction should also be rearranged to make the contents more logical. For instance, some literatures on gas-solid abrasive flow machining have been listed in the second paragraph of Introduction. Whereas, it seems that these is no any connection between these literatures.

2. In your introduction, are there any evidence to exhibit the unevenly processed surface after machined by AFM? Although Fu Y Z et al (2016) stated that there was a slight difference between the leading/trailing region than the blades’ center, mainly resulting from the irregular flow of abrasive media, it could be eliminated by redesigned the fixture. Hence, it doesn’t the strictly uneven finishing process. In addition, the abrasive media in their works was soft media and also simulated by standard k-ε turbulence model, so what’s the situation of abrasive media with polymer matrix?

3. Please highlights the novelty in your present work. Since there are numerous works concentrating on predicting the surface roughness or the material removal rate, so what’s the innovation difference between your present work with previous literatures.

4. In Figure 5, the title contents didn’t exhibit the pictures in order.

5. As you stated that AFM could make the surface uneven processed, how do you guarantee the machining uniformity in your work? I didn’t see any analysis or evidence in Section 2.

6. The Figure 3 exhibited the cloud diagram of collision contact force on the workpiece surface. While, how did you obtain these force cloud diagram? By simulation methods or just measured by some techniques. Please detailedly stated it in your article.

Author Response

Dear Editor:

The authors would like to thank the editor for providing valuable comments and suggestions on this manuscript. Our responses to the comments and suggestions are as follows and the manuscript was revised accordingly with the revised text highlighted in yellow.

Comment 1: The sentences in present articles needs to be carefully polished. Besides, the Abstract and Introduction should also be rearranged to make the contents more logical. For instance, some literatures on gas-solid abrasive flow machining have been listed in the second paragraph of Introduction. Whereas, it seems that these is no any connection between these literatures.

Response: The full text has been optimized, and the language of the Abstract and Introduction sections has been reorganized.

Comment 2: In your introduction, are there any evidence to exhibit the unevenly processed surface after machined by AFM? Although Fu Y Z et al (2016) stated that there was a slight difference between the leading/trailing region than the blades’ center, mainly resulting from the irregular flow of abrasive media, it could be eliminated by redesigned the fixture. Hence, it doesn’t the strictly uneven finishing process. In addition, the abrasive media in their works was soft media and also simulated by standard k-ε turbulence model, so what’s the situation of abrasive media with polymer matrix?

Response: Because the workpiece to be processed is a complex surface part, the pressure of the abrasive flow in contact with the workpiece is not uniform. Once the abrasive flow contacts the workpiece and friction wear occurs, the part that first contacts the abrasive flow will be subject to greater pressure, and there will be better polishing effect. As the abrasive flow continues to flow, the reduced pressure on the workpiece results in lower cutting forces, increasing surface roughness and uneven machining of the workpiece surface. Increasing the polishing time will make the surface quality of the workpiece more uniform, but the amount of material removal will increase, which does not meet the polishing requirements.

Comment 3: Please highlights the novelty in your present work. Since there are numerous works concentrating on predicting the surface roughness or the material removal rate, so what’s the innovation difference between your present work with previous literatures.

Response: A new surface finishing method is proposed, abrasive pool finishing. Through the combination of theory and experiment, the feasibility of the abrasive pool processing is verified each other, and the workpiece surface processing quality and the change of the workpiece surface roughness are used as the evaluation basis to explore the processing mechanism of the abrasive pool machining. According to the multiple linear regression equation, the abrasive pool is initially established Machining Surface Roughness Prediction Model.

Comment 4: In Figure 5, the title contents didn’t exhibit the pictures in order.

Response: Revised. (Line 159)

Comment 5: As you stated that AFM could make the surface uneven processed, how do you guarantee the machining uniformity in your work? I didn’t see any analysis or evidence in Section 2.

Response: Based on the advantages of various processing methods such as abrasive streamer finishing, magnetorheological finishing, and barrel finishing, as well as the fluid characteristics of gas-solid two-phase flow. In this paper, a new surface finishing method is proposed, that is, abrasive pool finishing. Mixing solid-phase abrasive with gas can obtain gas-solid two-phase flow with various flow states. Place the workpiece in a suitable gas-solid two-phase flow to make the abrasive contact, rub and collide with the surface of the workpiece, and use the abrasive as the cutter head to perform micro-cutting on the surface of the workpiece to realize the finishing of the workpiece surface by the gas-solid two-phase flow.

Comment 6: The Figure 3 exhibited the cloud diagram of collision contact force on the workpiece surface. While, how did you obtain these force cloud diagram? By simulation methods or just measured by some techniques. Please detailedly stated it in your article.

Response: Use EDEM software to simulate and analyze the collision between the abrasive particles and the workpiece to obtain a force cloud diagram, and calculate the removal amount of the workpiece material according to the collision force between the workpiece and the abrasive particles, which provides a reference for the subsequent processing time and other conditions. (Line 89-90)

Reviewer 3 Report

The ideas in the manuscript are very interesting, and the research results obtained have some potential applications of ceramic in the field of titanium alloy cutting. This manuscript is well structured and well written, which is easy to follow. The figures and tables are neat and easy to understand. The methodology is thoroughly explained and the work overall seems to be skillfully performed.

Generally, I think this manuscript can be accepted for publication after minor revision.

It is recommended to take notice the following:

A little bit of rearranging of the abstract is needed maybe. Abstract is usually divided into four parts WHY, WHAT, HOW, and main conclusions.

WHY  This section usually contains one or two lines mainly defining what is the objective of the study or this work was done.

WHAT  This is the main portion of the abstract. It contains what was done. Like what simulations have been performed what kind of parametric studies are done to support the WHY section.

HOW  In this section you will define how u have achieved the WHAT section points. What kind of methodologies you have utilized to achieve the goals defined in WHAT section?

Usually at the end you will include one or two lines that how it is going to benefit the scientific community or what are the readership of this paper.

Thus, the abstract of this manuscript should be modified.

It is recommended to add error bars in the Fig.6

Provide the values of R-squared and adjusted R-squared for your developed models. of course you can refer to the text in the section of 3.2 in the https://doi.org/10.1080/17480272.2022.2119432, and 3.1 in the https://doi.org/10.3390/f13091397.

Provide more information about the Range analysis.

L288: Ra should be italicized.

Author Response

Comment 3

Dear Editor:

The authors would like to thank the editor for providing valuable comments and suggestions on this manuscript. Our responses to the comments and suggestions are as follows and the manuscript was revised accordingly with the revised text highlighted in yellow.

Comment 1: A little bit of rearranging of the abstract is needed maybe. Abstract is usually divided into four parts WHY, WHAT, HOW, and main conclusions.

WHY  This section usually contains one or two lines mainly defining what is the objective of the study or this work was done.

WHAT  This is the main portion of the abstract. It contains what was done. Like what simulations have been performed what kind of parametric studies are done to support the WHY section?

HOW  In this section you will define how u have achieved the WHAT section points. What kind of methodologies you have utilized to achieve the goals defined in WHAT section?

Usually at the end you will include one or two lines that how it is going to benefit the scientific community or what are the readership of this paper.

Thus, the abstract of this manuscript should be modified.

Response: We have reorganized the language of the abstract as follows

The main purpose of this study is to explore a surface roughness prediction model of Gas-Solid Two-Phase Abrasive Flow Machining. In order to achieve the above purpose, an orthogonal ex-periment was carried out. Q235 steel as processing material and white corundum with different particle sizes as abrasive particles was used in experiment. Shape and spindle speed were the main reference factors. The range method and factor trend graph are used to comprehensively analyze the experimental results of different processing stages of the detection point, and the optimal parameter combination of A3B2C1D2 was obtained. According to the experimental results, a mul-tiple linear regression equation was established to predict the surface roughness, and the ex-perimental results were solved and significant analyzed by software to obtain a highly reliable prediction model. Through experiments, modeling and verification, it is known that the maximum average error between the obtained model and the actual value is 0.339 μm and the minimum is 0.008 μm, which can better predict the surface roughness of the gas-solid two-phase flow abrasive pool.

Comment 2: It is recommended to add error bars in the Fig.6

Provide the values of R-squared and adjusted R-squared for your developed models. of course you can refer to the text in the section of 3.2 in the https://doi.org/10.1080/17480272.2022.2119432, and 3.1 in the https://doi.org/10.3390/f13091397.

Provide more information about the Range analysis.

Response: The curve in this article uses the calculated average value. Since there are many curves in the figure, adding error bars will cause the whole picture to be very cluttered. Besides Figure 6 shows the average of five measurements roughness values of the same marked points of the round tube under the orthogonal experimental conditions (L168).

Comment 3: L288: Ra should be italicized.

Response: Revised

Round 2

Reviewer 2 Report

The paper has been well revised according to my suggestions.